# REGIONAL BASED QUERY IN GRAPH ACTIVE LEARNING

**Roy Abel**
Bar Ilan University
royabel10@gmail.com

**Yoram Louzoun**
Bar Ilan University
louzouy@math.biu.ac.il

## ABSTRACT

Graph convolution networks (GCN) have emerged as a leading method to classify nodes and graphs. These GCN have been combined with active learning (AL) methods when a small chosen set of tagged examples can be used. Most AL-GCN use the sample class uncertainty as selection criteria, and not the graph. In contrast, representative sampling uses the graph, but not the prediction. We propose to combine the two and query nodes based on the uncertainty of the graph around them. We here propose two novel methods to select optimal nodes in AL-GCN that explicitly use the graph information to query for optimal nodes. The first method named regional uncertainty is an extension of the classical entropy measure, but instead of sampling nodes with high entropy, we propose to sample nodes surrounded by nodes of different classes, or nodes with high ambiguity. The second method called *Adaptive Page-Rank* is an extension of the page-rank algorithm, where nodes that have a low probability of being reached by random walks from tagged nodes are selected. We show that the latter is optimal when the fraction of tagged nodes is low, and when this fraction grows to one over the average degree, the regional uncertainty performs better than all existing methods. While we have tested these methods on graphs, such methods can be extended to any classification problem, where a distance can be defined between the input samples.

## 1 INTRODUCTION

Relational information is often presented as graphs or multi-graphs, where nodes represent entities and edges represent relations between these entities. Such relations can be used to predict the class of the nodes, using two main principles. The first and most used method is based on node class homophily, where neighboring nodes belong to the same class with a high probability (Ji & Han, 2012; Berberidis & Giannakis, 2018; Zhu et al., 2003a;b; Sindhwani et al., 2005; Belkin & Niyogi, 2004). This has been used in many propagation-based algorithms where the class of a node is predicted using the class of neighboring nodes. The second approach presumes a correlation between the topological attributes (e.g. degree, centrality, clustering coefficient...) of nodes and their class (Shi & Malik, 2000; Yang et al., 2013; Rosen & Louzoun, 2015; Naaman et al., 2018; Cannistraci et al., 2013). These two principles are combined in Graph Convolutional Networks (GCN). Such networks received much interest over the last decade, and especially following the works of Kipf & Welling (2016), where they have produced higher accuracies than other label propagation methods. The main formalism proposed in such networks is the weighted combination of the input from previous layers in neighboring nodes:

$$X_{k+1} = \sigma(\widetilde{A} \times X_k \times W_k), \tag{1}$$

with $W_K$ being the weights of the $k$ layer, $X_k$ the input to this layer and $\widetilde{A}$ a matrix derived from the adjacency matrix (e.g. $\widetilde{A} = D^{-1/2}[A + I]D^{-1/2}$ for undirected graphs). $X_0$ is usually external

information about the nodes. In the absence of such information, the identity matrix (Schlichtkrull et al., 2018), topological features of nodes or the frequency of neighbors belonging to each class in the training set (Benami et al., 2019) have been proposed.

In the absence of a predefined set of classified nodes, and when the nodes composing the training set can be chosen, active learning can be used to query the class of nodes that would produce the highest precision (as defined through the prediction accuracy or any other measure on the entire dataset) using the minimal number of classified samples. Many Active Learning (AL) methods have been proposed (Lewis & Catlett, 1994; Culotta & McCallum, 2005; Settles & Craven, 2008). The most frequently used approaches are uncertainty sampling and representative sampling (Settles, 2009).

Uncertainty sampling is a general framework for measuring informativeness (Lewis & Catlett, 1994), where a learner queries the instance whose class is the most uncertain.Culotta & McCallum (2005) employed a simple uncertainty-based strategy for sequence models called least confidence (LC): $\phi^{LC}(x) = 1 - P(y^*|x; \Theta)$. Here, $y^*$ is the most likely label. This approach queries the instance for which the current model has the least confidence in its most probable label. Scheffer et al. (2001) proposed another uncertainty strategy, which queries the instance with the smallest margin between the posteriors for its two most probable labels: $\phi^M(x) = P(y_1^*|x; \Theta) - P(y_2^*|x; \Theta)$, where $y_1^*$ and $y_2^*$ are the first and second best labels, respectively. Another uncertainty-based measure of informativeness is entropy (see Shannon (1948)). For a discrete random variable Y , the entropy is given by $H(Y) = -\sum(P(y_i)log(P(y_i)))$. A different approach to uncertainty involves several independent models and looks for disagreement among them (Seung et al., 1992).

In representative sampling, one assumes that informative instances are "representative" of the underlying distribution, and the query is based on the properties of the nodes in contrast with the uncertainty sampling, where the predicted scores for each label are used and not the samples themselves. Measures of the distribution include the KullbackLeibler (KL) divergence similarity (McCallum & Nigam) or clustering (Xu et al., 2007), where the goal is to obtain representative labeled data samples. Fujii et al. (1998) considered a query strategy for nearest-neighbor methods that select queries that are (i) least similar to the labeled instances, and (ii) most similar to the unlabeled instances. Nguyen & Smeulders (2004) proposed a density-based approach that first clusters instances and then avoids querying outliers by propagating label information to instances in the same cluster. Settles and Craven (2008) suggested a general density-weighting technique combining both uncertainty and representative sampling. They query instances as follows: $\arg\max_X \phi_A(X) \times (\frac{1}{U} \sum_U sim(X, X_U))^\beta$ where $\phi_A(X)$ represents the informativeness of x according to some "base" query strategy A, and $U$ are the unlabeled samples. The second term weights the informativeness of x by its average similarity to all other instances in the input distribution (as approximated by $U$), subject to a parameter $\beta$ that controls the relative importance of the density term (Settles & Craven, 2008). Zhu et al. (2009) also proposed sampling by a combination of uncertainty and density to solve the outliers problem emerging in some uncertainty techniques.

Another frequently used measure is the influence of the unlabeled samples on the model, using varying methods, such as length of gradients (Settles et al., 2008), expected change or Fisher information ratio (Cohn et al., 1996). Finally, there are many hybrid methods that combine different criteria (Settles & Craven, 2008).

When applied to graphs, uncertainty methods were based on the properties of the classifier scores and did not explicitly use the graph information to select nodes. However, since neighboring nodes share classes more often than non-neighboring nodes, the graph itself can be used not only to predict the class of nodes but also to predict the diffusion of uncertainty. Similarly, representative sampling takes advantage of the graph but ignores the information on the nodes class. Assume, for example, two nodes: 1) One node that is a distinct connectivity component, where the current classifier gives the same score for both classes in binary classification. Checking the class of such a node, would probably not improve the accuracy of other nodes. 2) Similarly, a node with a predicted probability of 99 % to the first class would also not be of interest, even if this node has a very high degree, and checking its class would be of limited use since we already know it with a high probability. The interesting nodes to classify would be nodes that combine uncertainty and connection to many other nodes. We here propose two methods to combine uncertainty with graph properties and show

that combining the graph within the AL leads to significantly higher accuracies than all current AL methods on standard datasets.

## 2 RELATED WORK

Graphs have been extensively used for machine learning, especially in the context of GCN (Kipf & Welling, 2016; Schlichtkrull et al., 2018; Berg et al., 2017) and GNN (Grover & Leskovec, 2016; Rosen & Louzoun, 2015). GCNs were also used in combination with AL. However, as mentioned, in most such models, the graph is only used for the ML part, while the AL is performed ignoring the graph structure. Recently a few works have proposed to use the graphs themselves for AL. Three main types methods have been proposed: modularity, centrality, and label propagation:

In modularity approaches, nodes are divided into communities. Macskassy (2009) proposed to reveal the most central node in each community and sample it. Then each community is divided into sub-communities and the most central node in each sub-community is sampled and so on. Mackassy further suggested a hybrid method combining communities, centrality, and uncertainty with the Empirical Risk Minimization (ERM) framework. Ping et al. (2017) proposed combining community structure to perform batch-mode AL. They used communities to consider the overlap in information content among the "best" instances.

Centrality based approaches focus on "central" nodes using some method (e.g. high degree). The assumption is that the central nodes will have a major impact on the unknown labels. Macskassy (2009) in the ERM algorithm, showed that betweenness centrality is a good measure for centrality. Cai et al. (2017) proposed to calculate a node representativeness score based on graph centrality. They tested several centrality measures: degree centrality, betweenness centrality, harmonic centrality, closeness centrality, and page-rank centrality. They concluded that the Page-Rank centrality is superior, and suggested using it when the prediction model is not informative enough.

In label propagation approaches, the implicit assumption is of label smoothness over the graph or over the projection of the graph into some manifold in $R^N$. Ming Ji proposed to select the samples such that the total variance of the Gaussian field over unlabeled examples, as well as the expected prediction error of the harmonic Gaussian field classifier, is minimized. An efficient computation scheme was then proposed to solve the corresponding optimization problem with no additional parameter (Ji & Han, 2012).Ma et al. (2013) extended sub-modularity guarantees from V-optimality to $\Sigma$-optimality using properties specific to Gaussian Markov Random Field (GRMF)s.

Finally, Dimitris Berberidis proposed to sample the nodes with the highest expected change of the unknown labels. Thus, in contrast with the expected error reduction and entropy minimization approaches that actively sample with the goal of increasing the confidence on the model, Berberidis et al. focus on maximally perturbing the model with each node sampled. The intuition behind this approach is that by sampling nodes with the largest impact, one may take faster steps towards an increasingly accurate model (Berberidis & Giannakis, 2018).

## 3 MAIN CONTRIBUTIONS OF THE CURRENT WORK

We here propose to combine uncertainty and representative sampling methods. Specifically, we propose to sample within regions of uncertainty in the graph (in contrast with sampling uncertain nodes). While the sampling of nodes with high uncertainty was proposed in a vast array of applications (Cai et al., 2017; Zhou & Sun, 2014; Macskassy, 2009), sampling using regional uncertainty (and in the cases of directed graphs, regional directed uncertainty, as will be further explained) has never been successfully applied. Chen et al. recently proposed as one of their models, a regional uncertainty algorithm, but obtained very low accuracies (Chen et al., 2019). Specifically, three novel claims are proposed here:

- We propose that replacing the node entropy by the regional entropy leads to a higher accuracy in GCN based AL methods.
- We propose an extension of the PageRank algorithm to detect nodes with limited diffusion of information from labeled nodes in directed graphs, and show that an AL method based on this Adaptive PageRank (APR) outperforms other methods in some datasets.

- We hypothesize that information can be gained from the graph, only when the sampling rate is low enough so that most unlabeled nodes have no labeled networks. At this stage, we only exemplify this hypothesis but do not bring a clear proof for it.

## 4 MODEL AND DATA

### 4.1 DATA SETS

For our experiments, we used 6 real-world labeled networks including 4 citation network datasets: Cora, CiteSeer, PubMed (Sen et al., 2008), and an extended version of Cora denoted as Subelj Cora (kon, 2017), and 2 additional networks: Wikispeedia (West & Leskovec, 2012) and Email-Eu (Leskovec et al., 2007). In Cora and CiteSeer, each node has a bag of words (BOW) which is used as external features. Descriptions of all datasets, as well as statistics, can be found in Appendix A.1 and Table 3.

### 4.2 MACHINE LEARNING

Three different algorithms were tested for the class prediction: A) Random Forest (Breiman, 2001): 100 estimators, and a balanced class weight. B) XGBoost (Chen & Guestrin, 2016): Dart boosters, 15% internal validation, a max depth of 7, $\lambda = 1.3 \eta = 1.3, \gamma = 3$, a rate drop of 0.2, weighted sampling, softprob objective function, and early stopping after 10 steps. C) FFN - feed forward network, 2 hidden layers of sizes $F \times 2$ and $F \times 1/2$, where $F$ is the input dimension, learning rate 0.01, ReLU as activation function, drop out rate of 0.3, and $l_2$ penalty of 0.001, and 500 epochs with early stopping on 15% validation.

### 4.3 DIRECTED GCN

Given a graph $G = (V, E)$, with an adjacency matrix $A \in R^{N \times N}$ (binary or weighted), a diagonal degree matrix $D$, a node feature matrix $X_0 \in R^{N \times F}$ (i.e., F-dimensional feature vectors for each node), and a label vector $Y$, multi-layer GCN layers are defined as Eq. 1. The last layer is a soft-max used to determine the probability of each label. We use an extension to this model by incorporation of an asymmetric adjacency matrix (Benami et al., 2019).

We incorporate the direction by separating the adjacency matrix (asymmetric in directed graphs) into its symmetric and anti-symmetric components and concatenate them creating a $2n \times n$ adjacency matrix. The dimension of the output of each layer is: $[(2N \times N) \times (N \times i_n) \times (i_n \times o_n)] = 2N \times o_n$, which in turn is passed to the next layer following a rearrangement of the output by splitting and concatenating it to change dimensions from - $2N \times o_n$ to $N \times 2o_n$. A GCN model was run for 200 epochs with early stopping on 10% internal validation, with 1 hidden layer of size 16, learning rate of 0.01, ReLU as non-linear activation function, drop out rate of 0.6, weight decay of 0.005. Those hyperparameters are used for all data sets even though they were tuned to fit "Cora" data set. The tuning was performed using a regular training/test division and not in an AL setting.

### 4.4 INPUT VALUES

Some datasets had external information that could be used as input for the dataset. Others lacked such information. For datasets with no external information, topological input was used. Recent evidence suggests that nodes with similar classifications tend to have similar connection patterns in the network (Rosen & Louzoun, 2015; Cannistraci et al., 2013). Given a graph $G = (V, E)$, nodes were characterized by a large set of structural properties of local and global scale, composing a network attributes vector for each node. This embedding represented as a continuous vector in $R^F$ which can be used as an input for any classical machine learning algorithms, to determine the classes of the nodes. The following topological attributes were used for embedding: Attractor Basin (Muchnik et al., 2007), Average neighbor degree, Betweenness centrality (Brandes, 2008), BFS moments - first and second moments of the distance distribution obtained by BFS (Yoo et al., 2005), Closeness centrality (Sabidussi, 1966), Eccentricity, Fiedler vector (Ghosh & Boyd, 2006), Flow (Rosen & Louzoun, 2014), K-Core (Seidman, 1983), Louvain - community of the node (Blondel et al., 2008), Page Rank (Page et al., 1999), Motifs - number of sub-graphs which the node participates in for

each type of 3,4 motifs (Kashani et al., 2009). An alternative input was the number of first or second neighbors belonging to each class (Benami et al., 2019), as will be further discussed.

## 4.5 Existing Active Learning Methods For Comparison

We have compared the newly developed AL methods to state of the art AL methods. In all methods tested, we use a greedy approach where the nodes with the highest score (of the appropriate method) are selected. At each iteration, a batch of nodes is queried for their labels. Unless explicitly stated, the batch size is 1. The following existing AL methods have been tested: A) Centrality implemented using the Page Rank algorithm (Cai et al., 2017). B) Entropy - Shannon entropy (Shannon, 1948). C) Geo Dist - The shortest path length of all pairs of nodes with known and unknown classes is computed. The score is the minimum distance from a certain node to all known nodes. If there is no path connecting a node to any known node, a (high) score of 9 is given. D) Margin - the absolute difference between the probabilities of the two most likely classes (Scheffer et al., 2001). E) Rep Dist - Using the last layer output of the GCN model as a $R^C$ representation vector. Then choosing nodes that are different from labeled nodes to get representative distribution. We checked two distance definitions: MAH - Mahalanobis distance. LOF - Local Outlier Factor provided by "sklearn" package (Breunig et al., 2000). F) Feature AB - Attractor Basin (Muchnik et al., 2007) G) K Truss - Extension to k-core. $T_k$ is the largest subgraph such that all edges belong to at least $K - 2$ triangles. Node $v_i$ has a K-truss score $k$ if it belongs to the $k$ K-truss subgraph, but not to any $k + 1$ K-truss subgraph. Nodes with high K-Truss are argued to have a high influence on other nodes (Malliaros et al., 2016). K) Random - choose a node at random (i.e. no AL).

## 5 Regional AL Models

### 5.1 Regional Active Learning

Often, class labels are related in neighboring nodes. Thus, the uncertainty of the entire region can provide more information than the uncertainty of the node itself. Furthermore, observing the entire region can avoid outliers queries (e.g. nodes disconnected from the network), which is one of the main downsides of uncertainty sampling.

Specifically, we calculate for each node the expected probability of all classes using the current classifier. Then, for each node $i$ the regional probability is the average of the probabilities over the region

$$\tilde{p}(y_i = c) = \frac{1}{|N_i|} \sum_{j \in N_i} P(y_j = c) \tag{2}$$

where $N_i$ is the region (currently defined as first neighbors of the node $i$ in the undirected graph underlying the directed graph, but can be defined to be more distant or directed regions) and $y_j$ is the label of node $j$. This regional probability can be used as input to any uncertainty measure.

An alternative approach is the computation of the uncertainty measure for each node separately. In such a case, the regional uncertainty is calculated by averaging the score over the region (this approach denoted as AE):

$$\phi(v_i) = \frac{1}{|N_i|} \sum_{j \in N_i} \phi'(v_j) \tag{3}$$

where $\phi'(v_j)$ is a local uncertainty score of node $j$, and $\phi(v_i)$ is the regional uncertainty score of node $i$. This regional measure would give a high uncertainty to nodes close to many uncertain nodes (and not to nodes in heterogeneous regions), thus reducing uncertainty over wide areas speeding up the active learning process (see schematic Fig 1 for the difference). Both measures scale easily to large graphs, since calculating each node's region is required only once.

### 5.2 Adaptive PageRank - APR

PageRank has been proposed as a measure of centrality to choose informative nodes to query. However, PageRank scores rely only on the graph structure ignoring the nodes' labels. We here propose an adaptive extension of page rank, which also considers which nodes are labeled. PageRank (Page

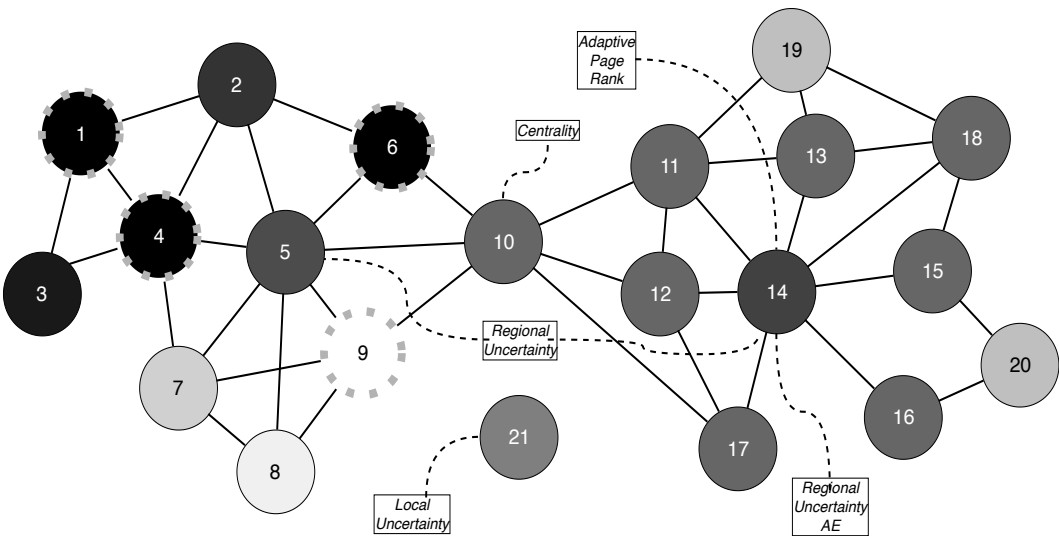

Figure 1: *Schematic figure: Binary classification task, where nodes can be either black or white. The label of nodes 1,4,6,9 (surrounded by dotted gray line) is already known. For each unlabeled node, the classifier prediction score is given by the grayscale between black and white. Local uncertainty techniques would query node 21. However, knowing the label of this node would have a minimal effect on other nodes. Regional uncertainty techniques would query the node with the most uncertain region. In this case, those are nodes 5 and 14, since the first is surrounded by nodes with high likelihood to both black and white (separates black and white), and the second is surrounded by nodes with high uncertainty. Standard centrality measures would query node 10, which is more central, ignoring the fact that this node is close to the two already known nodes (6,9), and thus probably would not add much information. In contrast, the adaptive approach which seeks the ratio between APR and PR would favor node 14, since it is far from the influence of nodes with known labels, thus will append more new information, and still is very central.*

et al., 1999) can be thought as a random walk process where each node receives an initial rank of 1, and at each iteration passes most of its rank through its out edges evenly. The rest is passed evenly to all nodes in the graph. The Adaptive Page Rank (APR) follows the same concept, but the initial rank of each node is 1 if the class label is known and 0 otherwise. Then, for each node, a random node is chosen with a probability of $\gamma$, and one of the nodes' neighbors is chosen with a probability of $1 - \gamma$, and the rank is passed to the chosen node. This process is repeated until convergence is obtained. The Rank of nodes with known class is fixed to be 1 across all iterations.

The steady state of APR (the vector of APR values over all nodes) is obtained when:

$$APR = \gamma(A_N^T \times APR) + (1 - \gamma) \tag{4}$$

where APR is a $R^N$ vector of scores, $\gamma \in (0, 1)$, and $A_N = D^{-1}A$. Since the values of APR are fixed for the labeled nodes, we can solve the equation above through:

$$\begin{aligned} APR(U) &= \gamma A_N[U, L]^T \times APR(L) \\ &+ \gamma A_N[U, U]^T \times APR(U) + 1 - \gamma, \end{aligned} \tag{5}$$

where $U$ represents the indexes of the unknown nodes and $L$ represents the labeled nodes, leading to:

$$APR(U) = (I - \gamma A_N[U, U]^T)^{-1}(\gamma A_N[U, L]^T \times APR(L) + 1 - \gamma) \tag{6}$$

The APR score can be described as the influence of the labeled nodes over the unknown nodes. One can use this measure in an AL framework seeking the best ratio between APR and PR to choose nodes, which on the one hand are not affected by the known nodes, and on the other hand, would affect the other nodes (See Fig 1). In the following table, we denote the APR to PR ration as the APR method.

Table 1:    *Results (with BOW) -Average accuracy on unlabeled nodes for Cora and CiteSeer networks, with bag of words as input. Those results obtained with 20 X number of classes labeled nodes, when the initial state was 4 random nodes from each class (same as Cai et al. (2017))*
*\* score is estimated from figure        \*\* scores for smaller budget, since it was the max reported*

| algorithm | Cora | CiteSeer |
|---|---|---|
| *Random* | 0.803 | 0.697 |
| Kipf (random) (Kipf & Welling, 2016) | 0.801 | 0.679 |
| Kipf (specific train) (Kipf & Welling, 2016) | 0.815 | 0.703 |
| Chang const params (Cai et al., 2017) | 0.823 | 0.707 |
| Chang adaptive params (Cai et al., 2017) | 0.825 | **0.721** |
| HNE (Chen et al., 2019) | 0.59* | - |
| TV/MSD** (Berberidis & Giannakis, 2018) | 0.78* | 0.7* |
| ClassSeek** (McDowell, 2015) | 0.805* | 0.695* |
| Σ-Opt** (Ma et al., 2013) | 0.73* | 0.71* |
| ALFNET (Bilgic et al., 2010) | 0.78* | 0.7* |
| Geo Dist | 0.806 | 0.680 |
| Attractor Basin (Muchnik et al., 2007) | 0.585 | 0.646 |
| K Truss (Malliaros et al., 2016) | 0.709 | 0.693 |
| rep dist LOF | 0.803 | 0.691 |
| rep dist MAH | 0.803 | 0.691 |
| PR | 0.802 | 0.691 |
| APR | 0.815 | 0.693 |
| entropy | 0.803 | 0.696 |
| region entropy | 0.814 | 0.691 |
| region entropy AE | 0.821 | 0.693 |
| margin | 0.800 | 0.700 |
| region margin | **0.827** | 0.705 |
| region margin AE | 0.818 | 0.675 |

## 6    RESULTS

In order to test the effect of different AL schemes, we first tested the best machine learning framework for the inference task in three of the datasets studied here (Table 3). We have tested four algorithms: XGBoost, FFN, GCN, and RF (see Model And Data). For each algorithm, we have tested three types of inputs: A) Topological features - a set of node topological features, such as their degree, the clustering coefficient, and the frequency of subgraphs frequencies, as proposed by Benami et al. (2019) (see Model And Data for the full features). B) Neighbors training set class label sum. We summed for each node the number of neighbors belonging to each class in the training set. The sum was represented as a vector of sums (e.g. if a node has 10 neighbors, only three of which are in the training set, with two belonging to the first class, and one belonging to the third class, the vector would be $[2, 0, 1, ..]$). The sum was performed on first and second neighbors producing a vector of twice the number of classes. C) A combination of the two as a concatenated version.

The precision was computed for different training set fractions in a passive set-up where the training set is pre-defined. In the vast majority of datasets studied, a GCN with the number of neighbors in the training set belonging to each class obtained the highest Micro and Macro F1 scores (Bold line in Appendix Fig 2). We, thus, only used this set-up for all the AL schemes.

### 6.1    ACTIVE LEARNING RESULTS

We here propose that in graph-based AL the combination of uncertainty and representation through regional information leads to higher accuracies than each by itself. We first tested multiple existing local AL approaches (see Model And Data). For each dataset studied, we computed the accuracy of the model, as well as its loss and micro and macro F1 scores for the different AL approaches, as a function of the number of nodes queried (Appendix Fig 3). We queried 1 node at a time, except for the Pubmed and Subelj datasets, where we used a batch size of 5. In the Cora and CiteSeer datasets,

BOW information was also available. We thus tested in these two datasets either a neighbor-class based classification (i.e. classification using no external content, only the labels of the neighbors around each node) or a BOW based classification. In the latter, we stopped after 200 classified nodes.

Many AL schemes actually perform worse than random (table 2, and Appendix Fig 3). On average the entropy produces the best accuracy for different datasets for most sampling rates and for the last time point. No major differences were detected between the micro and macro F1. We thus report only accuracy results. The F1 score and accuracies were computed using the default setting of the F1 score of Pedregosa et al. (2011). All simulations were repeated 20 times. We do not plot standard errors to avoid cluttering the figures, but the average standard error was less than 0.015 and is thus much smaller than the difference between typical methods.

In the regional AL schemes, The analysis was similar to the local AL schemes with the same number of simulations and setup. In the Cora, CiteSeer and PubMEd datasets the APR gives the best results for low sampling rates, and then the region entropy gives better results for higher sampling rates. In the Email-EU and Wikispedia the AE region entropy is the best method for most sampling rates (in the 15%, the Geo-dist outperforms them in Email-EU). In the SubjelCora, the Region Margin is the best result.

We have also compared the performance of the different algorithms proposed here to existing performances in the standard Cora and CiteSeer datasets (see Table 1 for the results obtained with BOW). We outperform all existing methods in the Cora dataset and get high accuracies, yet as high as the ones obtained by Cai et al methods, with much simpler algorithms in the CiteSeer dataset.

While APR outperforms all other methods at low sampling fractions (typically less than 5 %) in some datasets (Cora, CiteSeer and PubMed), it never outperforms the regional uncertainty methods at high fractions (15 %). A putative explanation for that may be that at high sampling fraction, Almost all nodes in the graph neighbor sampled nodes. In such cases, uncertainty may be more important than the distance to sampled nodes. To check that we computed the distance of a typical node to a randomly sampled node as a function of the sampling fraction. This distance drops to a plateau around 5 % sampling (see Appendix Fig 4) suggesting that beyond this fraction, sampling more nodes does not provide a significant regional advantage. We suggest this conjecture as a general method to estimate the sampling range where regional methods are advantageous and now plan to study it in detail.

## 7  CONCLUSIONS

We have here shown that the accuracy of AL when uncertainty is computed regionally is much higher than when either local uncertainty or representative nodes are used in most sets (with the exception of CiteSeer with the BOW). When computing regional uncertainty, two flavors were proposed. Either computing the uncertainty on the average class probability of neighbors or computing the average uncertainty in neighbors (marked as AE). The first is better with a low number of classes or low degree. The later (AE) is better when the number of classes and the degree are high (e.g. Email-EU or Wikispeedia). In high class number/degree, the average class probability approaches the mean-field and is of limited interest. Furthermore, sampling depths can be divided into two main phases. In the most early phase, when the fraction of the tagged nodes is very small (much smaller than one over the average degree), the best nodes to query are nodes that can maximize the propagation of information into yet unstudied regions. As the graphs are more densely sampled, a regional approach is optimal, where the best nodes to query are those in regions of high uncertainty (e.g. the regional entropy approach proposed here). At such intermediate tagged fractions, the main goal of querying the oracle is to find enough samples in the vicinity of each node with high uncertainty. Beyond such a fraction, only the node itself is of importance, and graph independent approaches can be used.

While we cannot provide a deterministic predictive method on what approach should be used, we propose the two following rules of thumb. If the number of classes and the average degree are small (e.g. Pubmed or Cora), either region entropy or region margin can be used with no clear advantage between them. In the same conditions, at very low sampling depth, APR is often the best method. If the number of classes or degree are high (e.g. Wikipeedia), the AE version of the same should be used.

Table 2: *Results - accuracy without content - Average prediction results (accuracy) on all unlabeled nodes, with neighbors labels features as input. The results reported for different fractions of labeled nodes, when the initial state was 1 random nodes from each class. In contrast with Table 1, here no external information on the nodes (such as a BOW) was used.*

| algorithm | cora | | | CiteSeer | | | PubMed | | | |
|---|---|---|---|---|---|---|---|---|---|---|
| | 5% | 10% | 15% | 5% | 10% | 15% | 3% | 5% | 10% | 15% |
| Random | 0.668 | 0.731 | 0.766 | 0.442 | 0.509 | 0.547 | 0.699 | 0.731 | 0.765 | 0.789 |
| Geo Dist | 0.616 | 0.710 | 0.763 | 0.456 | 0.526 | 0.561 | 0.671 | 0.701 | 0.751 | 0.775 |
| Rep Dist MAH | 0.566 | 0.642 | 0.690 | 0.319 | 0.345 | 0.358 | 0.683 | 0.716 | 0.745 | 0.783 |
| Rep Dist LOF | 0.468 | 0.587 | 0.633 | 0.338 | 0.357 | 0.359 | 0.680 | 0.730 | 0.780 | 0.809 |
| Entropy | 0.676 | 0.766 | 0.820 | 0.433 | 0.551 | 0.608 | 0.692 | 0.739 | 0.779 | 0.807 |
| Margin | 0.256 | 0.269 | 0.391 | 0.300 | 0.371 | 0.428 | 0.482 | 0.512 | 0.467 | 0.479 |
| PR | 0.676 | 0.726 | 0.768 | 0.451 | 0.525 | 0.553 | 0.687 | 0.722 | 0.760 | 0.781 |
| | | | | | | | | | | |
| Region Entropy | 0.676 | 0.767 | 0.825 | 0.454 | 0.554 | **0.616** | 0.704 | **0.745** | **0.789** | **0.826** |
| Reg Entropy AE | 0.590 | 0.753 | **0.834** | 0.401 | 0.501 | 0.571 | 0.508 | 0.606 | 0.653 | 0.742 |
| Region Margin | 0.668 | 0.749 | 0.806 | 0.458 | 0.518 | 0.551 | 0.678 | 0.739 | 0.771 | 0.824 |
| Reg Margin AE | 0.626 | **0.775** | 0.827 | 0.439 | 0.516 | 0.583 | 0.658 | 0.730 | 0.772 | 0.790 |
| APR | **0.700** | 0.753 | 0.784 | **0.465** | **0.564** | 0.607 | **0.728** | 0.733 | 0.771 | 0.781 |

| algorithm | Email-EU | | | Wikispeedia | | | SubeljCora | | | |
|---|---|---|---|---|---|---|---|---|---|---|
| | 5% | 10% | 15% | 5% | 10% | 15% | 3% | 5% | 10% | 15% |
| Random | 0.427 | 0.484 | 0.470 | 0.500 | 0.501 | 0.496 | 0.678 | 0.710 | 0.749 | 0.772 |
| Geo Dist | 0.389 | 0.504 | **0.586** | 0.494 | 0.526 | 0.532 | 0.502 | 0.549 | 0.635 | 0.694 |
| Rep Dist MAH | 0.388 | 0.460 | 0.476 | 0.398 | 0.419 | 0.407 | 0.649 | 0.680 | 0.718 | 0.742 |
| Rep Dist LOF | 0.338 | 0.349 | 0.360 | 0.407 | 0.437 | 0.401 | 0.637 | 0.683 | 0.720 | 0.753 |
| Entropy | 0.418 | 0.493 | 0.525 | 0.485 | 0.540 | 0.538 | 0.671 | 0.716 | 0.767 | 0.799 |
| Margin | 0.367 | 0.452 | 0.534 | 0.382 | 0.345 | 0.347 | 0.300 | 0.486 | 0.668 | 0.736 |
| PR | 0.301 | 0.174 | 0.310 | 0.336 | 0.325 | 0.351 | 0.660 | 0.694 | 0.734 | 0.760 |
| | | | | | | | | | | |
| Region Entropy | 0.235 | 0.321 | 0.328 | 0.327 | 0.375 | 0.376 | 0.631 | 0.674 | 0.768 | 0.807 |
| Reg Entropy AE | **0.451** | **0.523** | 0.570 | **0.533** | **0.558** | **0.566** | 0.536 | 0.628 | 0.754 | 0.806 |
| Region Margin | 0.385 | 0.396 | 0.353 | 0.494 | 0.476 | 0.406 | **0.687** | **0.727** | **0.788** | **0.819** |
| Reg Margin AE | 0.387 | 0.486 | 0.567 | 0.510 | 0.548 | 0.564 | 0.544 | 0.647 | 0.729 | 0.799 |
| APR | 0.382 | 0.491 | 0.554 | 0.452 | 0.495 | 0.495 | 0.665 | 0.698 | 0.728 | 0.753 |

There are many active learning approaches including uncertainty, representative, influence, error reduction. We have shown, as have others before (Settles & Craven, 2008; Zhu et al., 2009; Macskassy, 2009; Cai et al., 2017) that single measure methods often do not produce a gain extending to high sampling fraction compared with random sampling. Thus, hybrid techniques, combining several approaches, outperform using only one approach have been proposed (Cai et al., 2017). We here proposed two novel measures, regional uncertainty, and adaptive page rank, which are themselves hybrid methods between uncertainty and representation. Those two measures can be further combined with other existing active learning techniques to achieve even better performance than random sampling. We have shown that these methods can be used with or without external information on the nodes.

While we have studied here only graph-based ML, the same approach can be used in any task where a distance metric can be defined on the input samples. In such cases, the region of each node can be defined using distances instead of edges.

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

# A  APPENDIX

## A.1  DATA SETS

### A.1.1  CORA

The Cora dataset (McCallum et al., 2000) consists of 2,708 scientific machine learning publications categorized into one of seven topics. The citation network consists of 5,429 edges. We consider only papers which are cited by or cite other papers. Bag of words (BOW) is also available for each node. We used it as in the original publication (Sen et al., 2008).

### A.1.2  CITESEER FOR DOCUMENT CLASSIFICATION

The CiteSeer data set (Giles et al., 1998) consists of 3,312 scientific publications classified into six categories: Agents, Artificial Intelligence, Database, Human Computer Interaction, Machine Learning, and Information Retrieval. There are 4,732 edges describing citations in the data set. Again, BOW is also available for each node and is used as in the original publication (Sen et al., 2008).

### A.1.3  EMAIL-EU-CORE NETWORK

The Email-Eu-core network (Leskovec et al., 2007; Leskovec & Krevl, 2014; Yin et al., 2017) was generated using email data from a large European research institution. There are 25,571 edges (u,v) in the network representing at least one email from a person $u$ to a person $v$. The emails only represent communications between institution members core of 1,005 people. The dataset also contains "ground-truth" community memberships of the nodes. Each individual belongs to exactly one of 42 departments at the research institute. This network represents the "core" of the email-EuAll network, which also contains edges between members of the institution and people outside of the institution.

### A.1.4  PUBMED DIABETES

The PubMed Diabetes dataset consists of 19,717 scientific publications from the PubMed database pertaining to diabetes classified into one of three classes. The citation network consists of 44,338 edges. BOW is also available, and each publication in the dataset is described by a TF/IDF weighted word vector from a dictionary which consists of 500 unique words (Sen et al., 2008).

### A.1.5  SUBELJ CORA

Citation network of 23,166 scientific computer science publications classified into one of ten categories: Artificial Intelligence, Operating Systems, Data Structures Algorithms and Theory, Programming, Networking, Encryption and Compression, Human Computer Interaction, Databases, Hardware and Architecture, Information Retrieval. The citation network consists of 91,500 edges indicating that the left node cited the right node (kon, 2017).

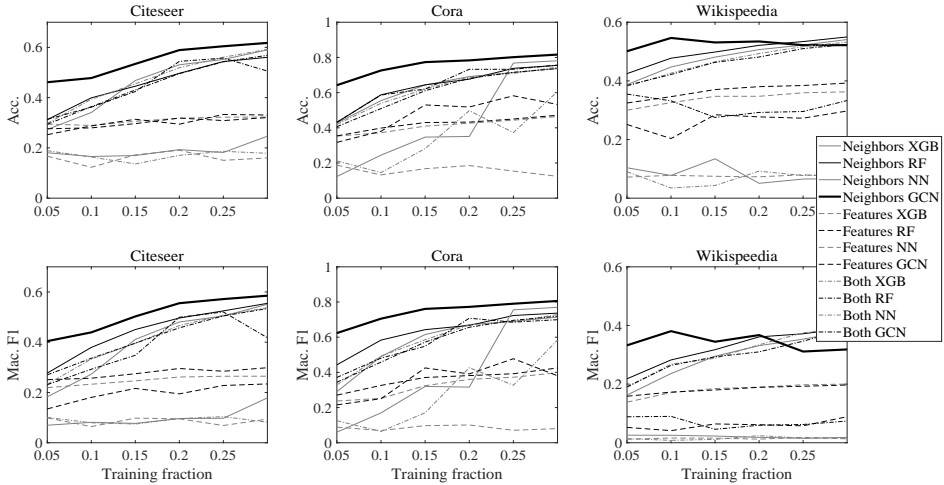

Figure 2: *Accuracy and Macro F1 as a function of training set fraction in 3 datasets for different learning methods. We tested for three reported datasets multiple precision estimates as a function of the training set fraction. We have tested four algorithms: GXBoost, FFN, GCN and RF. For each algorithm, we tested three types of input, the neighbors class, topological features of the node and the combination of the two. The precision was computed for different training set fractions in a passive setup where the training set is pre-defined. One can clearly see that in all datasets and using all measures, the GCN with the neighbors class as input produces the best accuracies (black thick line).*

### A.1.6 WIKISPEEDIA NAVIGATION PATHS

This dataset is collected from the human-computation game Wikispeedia. In Wikispeedia, users are asked to navigate from a given source to a given target article, by only clicking Wikipedia edges. A condensed version of Wikipedia is used, with 4,604 articles, and 119,882 directed edges connecting them.

Each article is classified by its subject into one of the following: History, People, Countries, Geography, Business Studies, Science, Everyday Life, Design and Technology, Music, IT, Language and Literature, Mathematics, Religion, Art, Citizenship (West & Leskovec, 2012; West et al., 2009).

Table 3: Dataset statistics

| Data Set | Nodes | Edges | Classes |
|---|---|---|---|
| CORA | 2,708 | 5,429 | 7 |
| CITESEER | 3,312 | 4,732 | 6 |
| EMAIL-EU | 1,005 | 25,571 | 42 |
| PUBMED | 19,717 | 44,338 | 3 |
| SUBELJ CORA | 23,166 | 91,500 | 10 |
| WIKISPEEDIA | 4,604 | 119,882 | 15 |

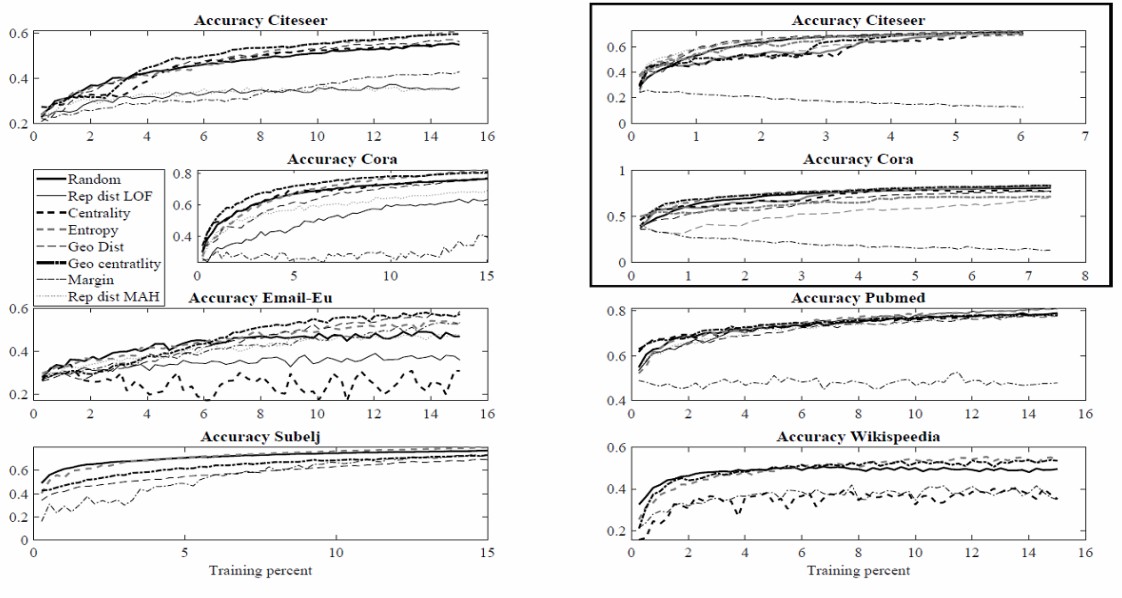

Figure 3: *Accuracy as a function of sampling fraction for different datasets and different local AL methods. Results lower than the random sampling line (Thick gray line) represent AL algorithms that do not contribute to the accuracy. The two subplots surrounded by a box are the results with the BOW. We have tested Micro and Macro F1 as well as the loss, with similar results (data not shown).*

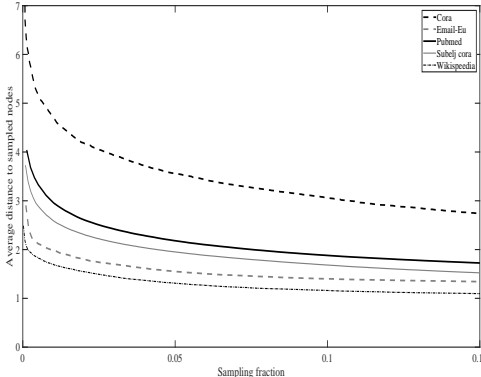

Figure 4: *Distance to randomly sampled nodes as a function of the fraction of sampled fraction. Within a sampling fraction of 5 %, the average distance to a sampled node reaches a fixed distance. The distance was computed, by setting a constant fraction (x axis) of random nodes to be marked, and measuring the average distance of unmarked nodes to the closes marked node. (y axis). In CiteSeer is the same but the distance is larger.*

