# OpenReview forum: "Regional based query in graph active learning"
_ICLR.cc/2020/Conference — Reject_

### Official Review · AnonReviewer1 · 2019-10-23
**Official Blind Review #1**

**Rating:** 1

**Review:**

The paper studies active learning in graph representations. To decide which nodes in a graph to label during the active labeling, the paper proposes two approaches. First it argues that one should consider region-based measures rather than single nodes. Second, it proposes to adapt the page rank algorithm (APR) to determine which nodes are far away from labeled nodes.

Overall it remains unclear *how* to select the right strategy (before seeing the results for a dataset) i.e. which of the proposed approaches or variants should one select for a new dataset.

Strength:
-	One of the ideas of the paper, using region entropy over single node entropy makes sense to me.
-	The paper evaluates on 6 datasets and compares different variants as well to related work on 2 datasets.

Weaknesses:
1.	The paper contains several confusing and contradicting statements or claims which are not supported by the experimental results:
For example:
1.1.	“APR outperforms all other methods at low sampling fractions”.  This is supported neither in Table 1 nor Table 2, where APR is frequently not highest performing
1.2.	 “We have here shown that the accuracy of AL when uncertainty is computed regionally is much higher than when either local uncertainty or representative nodes are used”, this is not the case on CiteSeer in Table 1
1.2.1.	Also e.g. “Region Margin” is worse than random on 5% Email-EU; or “Region Margin AE” on 3% SubeljCora (Table 2) [It is unclear how to select with or without AE]
1.3.	“We outperform all existing methods in the Cora dataset, and get very similar results to the best accuracy obtained by Chang et al methods:”
1.3.1.	The difference to Cai et al. on Cora is very small (improvement by only 0.002), while on Citeseer the performance is comparatively bigger (Cai et al. is by 0.016 better)
1.3.2.	It should be “Cai et al”
2.	Clarity: I found the paper rather difficult to understand and follow:
Some specifics:
2.1.	The introduction could be more concisely discussing the motivation, the main idea of the paper, as well as contributions.
2.2.	Figure 1: according to the caption, APR should point to node 15, but in the figure it points to node 14. From the example it makes much more sense to label node 14 to me.
2.3.	Page 6 mentions twice the “ratio between APR and PR”, is this is this used/evaluated in the results?
2.4.	The decision what is bold and what is not is not consistent throughout the table 2.
2.5.	“Thus, hybrid techniques, combining several approaches, outperform using only one approach have been proposed.” It is not clear what this refers to and where the hybrid techniques have been evaluated.

Minor:
The paper contains many minor writing issues, e.g.
-	missing spaces, e.g. “distribution,and” (page 2)
-	Table 1: incomplete sentence: “∗∗ scores for smaller budget, since it was the”
-	Table 2: unclear: “accuracy without content”

The paper’s incorrect claims (weakness 1) are highly concerning and strongly suggest rejecting the paper. Furthermore, the clarity of the paper should be improved to follow the author arguments and make the paper easier to read.



**Experience Assessment:**

I have published one or two papers in this area.

**Review Assessment: Checking Correctness Of Derivations And Theory:**

I assessed the sensibility of the derivations and theory.

**Review Assessment: Checking Correctness Of Experiments:**

I carefully checked the experiments.

**Review Assessment: Thoroughness In Paper Reading:**

I read the paper thoroughly.

---

> ### Author Response · Authors · 2019-11-13
> **Answer to reviewer 2 clarity and wording issues.**
>
> Reviewer 2 has the same main comment as reviewer 1. “Overall it remains unclear *how* to select the right strategy (before seeing the results for a dataset) i.e. which of the proposed approaches or variants should one select for a new dataset.”. Again, we added now such a section in the discussion.
>
> Beyond that the main critiques of reviewer 2 are on clarity and some minor wording issues. We have now edited all the issues mentioned by the reviewer, as well as other aspects of the paper that required extra clarity.
>
> Following is a detailed answer to the reviewer
>
> Review: The paper studies active learning in graph representations. To decide which nodes in a graph to label during the active labeling, the paper proposes two approaches. First it argues that one should consider region-based measures rather than single nodes. Second, it proposes to adapt the page rank algorithm (APR) to determine which nodes are far away from labeled nodes.
>
> Overall it remains unclear *how* to select the right strategy (before seeing the results for a dataset) i.e. which of the proposed approaches or variants should one select for a new dataset. Strength: - One of the ideas of the paper, using region entropy over single node entropy makes sense to me. - The paper evaluates on 6 datasets and compares different variants as well to related work on 2 datasets. Weaknesses:
> 1. The paper contains several confusing and contradicting statements or claims which are not supported by the experimental results: For example:
> 1.1. “APR outperforms all other methods at low sampling fractions”. This is supported neither in Table 1 nor Table 2, where APR is frequently not highest performing
>
> Answer. This is only half the statement. The full statement was “As mentioned in multiple datasets, the APR outperforms all other methods at low sampling fractions(typically less than 5 %).”.  We were explicit about the fact that this was in only some dataset. We have now further clarified that in case this was not clear enough.
>
> 1.2. “We have here shown that the accuracy of AL when uncertainty is computed regionally is much higher than when either local uncertainty or representative nodes are used”, this is not the case on CiteSeer in Table 1
>
>
> Answer.  Indeed, but it is true in all other datasets, for most sampling sizes. We have stated that as a general statement, and have now clarified it to avoid any possible doubt.
>
> 1.2.1. Also e.g. “Region Margin” is worse than random on 5% Email-EU; or “Region Margin AE” on 3% SubeljCora (Table 2) [It is unclear how to select with or without AE]
>
>
> Answer. We now discuss clearly model choice.
>
> 1.3. “We outperform all existing methods in the Cora dataset, and get very similar results to the best accuracy obtained by Chang et al methods:”
>  1.3.1. The difference to Cai et al. on Cora is very small (improvement by only 0.002), while on Citeseer the performance is comparatively bigger (Cai et al. is by 0.016 better) 1.3.2. It should be “Cai et al”
>
>
> Answer. The statement was rephrased and the reference was corrected too.
>
> Clarity: I found the paper rather difficult to understand and follow: Some specifics: 2.1. The introduction could be more concisely discussing the motivation, the main idea of the paper, as well as contributions. 2.2. Figure 1: according to the caption, APR should point to node 15, but in the figure it points to node 14. From the example it makes much more sense to label node 14 to me.
>
> Answer. This was indeed a typo and it was corrected.
>
> 2.3. Page 6 mentions twice the “ratio between APR and PR”, is this is this used/evaluated in the results?
>
> Answer. The ratio between APR and PR is precisely what is used. This is now clarified in the text.
>
> 2.4. The decision what is bold and what is not is not consistent throughout the table 2. 2.5.
>
> Answer. This was also corrected.
>
> “Thus, hybrid techniques, combining several approaches, outperform using only one approach have been proposed.” It is not clear what this refers to and where the hybrid techniques have been evaluated.
>
> Answer. This referred to previous work and is now clarified.
>
> Minor: The paper contains many minor writing issues, e.g. - missing spaces, e.g. “distribution,and” (page 2) - Table 1: incomplete sentence: “∗∗ scores for smaller budget, since it was the” - Table 2: unclear: “accuracy without content”
>
> Answer. These minor issues were corrected.
>
> The paper’s incorrect claims (weakness 1) are highly concerning and strongly suggest rejecting the paper. Furthermore, the clarity of the paper should be improved to follow the author arguments and make the paper easier to read.
>
> As explained the claim that the reviewer claims to be incorrect was half a sentence, when the entire sentence is read, there were no incorrect claims. We would thus appreciate a change of decision by the reviewer.

---

### Official Review · AnonReviewer3 · 2019-10-24
**Official Blind Review #3**

**Rating:** 6

**Review:**

The authors present an algorithm for actively learning the nodes in a graph that should be sampled/labeled in order to improve classifier performance the most. The proposed techniques use both the graph structure, and the current classifier performance/accuracy into account while (actively) selecting the next node to be labeled.

There seem to be two main contributions in the paper. 1) The propose to sample nodes nodes based on "regional" uncertainty rather than node uncertainty 2) They use an variant of pagerank to determine nodes that are central, and hence most likely to affect subsequent classification in graph convolution classifiers. Both approaches seem to be interesting. There are experiments to show effectiveness of these techniques, and there are some interesting observations (for example, that the APR technique works better for smaller sample sizes, while the regional uncertainty methods do better for larger sampling fractions.).

While both techniques seem straightforward extensions of previous approaches (and are well explained in the paper),  the experiments indicate that they work better than prior approaches. It would have been nice if the authors had also discussed ways in which one or more of these techniques could be combined though, or discussed how we could pick the right approach (in a more empirical way, since it is not clear what the threshold for high sampling rate/low sampling rate distinction is, or if it varies from problem to problem)

**Experience Assessment:**

I do not know much about this area.

**Review Assessment: Checking Correctness Of Derivations And Theory:**

N/A

**Review Assessment: Checking Correctness Of Experiments:**

I assessed the sensibility of the experiments.

**Review Assessment: Thoroughness In Paper Reading:**

I read the paper at least twice and used my best judgement in assessing the paper.

---

> ### Author Response · Authors · 2019-11-13
> **How to select model.**
>
> Reviewer 1 has one main critique, which is that there is no clear explanation of how to choose in advance a method. This critique is also shared by the second reviewer. Following both comments reviews, we now add a clear section on how to choose among the different algorithms that we propose.

---

### Decision · Program_Chairs · 2019-12-19

**Decision:**

Reject

**Comment:**

The paper proposes a method for performing active learning on graph convolutional networks. In particular, instead of performing uncertainty-based sampling based on an individual node level, the authors propose to look at regional based uncertainty. They propose an efficient algorithm based on page rank. Empirically, they compare their method to several other leading methods, comparing favorably.

Reviewers found the work poorly organized and difficult to read. The idea to use region based estimates is intuitive but feels like nothing more than just that. It's not clear if there is a mathematical basis to justify such a method (e.g. an analysis of sample complexity as has been accomplished in other graph active learning problems, Dasarathy, Nowak, Zhu 2015).

The idea requires further study and justification, and the paper needs an improved exposition. Finally, the authors were not anonymized on the PDF.